# An Integrative Glycomic Approach for Quantitative Meat Species Profiling

**DOI:** 10.3390/foods11131952

**Published:** 2022-06-30

**Authors:** Sean Chia, Gavin Teo, Shi Jie Tay, Larry Sai Weng Loo, Corrine Wan, Lyn Chiin Sim, Hanry Yu, Ian Walsh, Kuin Tian Pang

**Affiliations:** 1Bioprocessing Technology Institute, Agency for Science Technology and Research (A*STAR), Singapore 138668, Singapore; sean_chia@bti.a-star.edu.sg (S.C.); gavin_teo@bti.a-star.edu.sg (G.T.); tay_shi_jie@bti.a-star.edu.sg (S.J.T.); corrine_wan@bti.a-star.edu.sg (C.W.); sim_lyn_chiin@bti.a-star.edu.sg (L.C.S.); ian_walsh@bti.a-star.edu.sg (I.W.); 2Institute of Bioengineering and Bioimaging, Agency for Science Technology and Research (A*STAR), Singapore 138669, Singapore; larry_loo@ibb.a-star.edu.sg (L.S.W.L.); hyu@ibb.a-star.edu.sg (H.Y.); 3Department of Physiology, The Institute for Digital Medicine (WisDM), Yong Loo Lin School of Medicine, National University of Singapore, Singapore 117593, Singapore; 4Mechanobiology Institute, National University of Singapore, Singapore 117411, Singapore; 5CAMP, Singapore-MIT Alliance for Research and Technology, Singapore 138602, Singapore

**Keywords:** O-glycan, N-glycan, glycomic, meat species

## Abstract

It is estimated that food fraud, where meat from different species is deceitfully labelled or contaminated, has cost the global food industry around USD 6.2 to USD 40 billion annually. To overcome this problem, novel and robust quantitative methods are needed to accurately characterise and profile meat samples. In this study, we use a glycomic approach for the profiling of meat from different species. This involves an O-glycan analysis using LC-MS qTOF, and an N-glycan analysis using a high-resolution non-targeted ultra-performance liquid chromatography-fluorescence-mass spectrometry (UPLC-FLR-MS) on chicken, pork, and beef meat samples. Our integrated glycomic approach reveals the distinct glycan profile of chicken, pork, and beef samples; glycosylation attributes such as fucosylation, sialylation, galactosylation, high mannose, α-galactose, Neu5Gc, and Neu5Ac are significantly different between meat from different species. The multi-attribute data consisting of the abundance of each O-glycan and N-glycan structure allows a clear separation between meat from different species through principal component analysis. Altogether, we have successfully demonstrated the use of a glycomics-based workflow to extract multi-attribute data from O-glycan and N-glycan analysis for meat profiling. This established glycoanalytical methodology could be extended to other high-value biotechnology industries for product authentication.

## 1. Introduction

With the growing human population and the increasing demand for food, food adulteration has become a global problem estimated to affect 10–20% of all food consumed in the world [1,2]. Such contamination by either additions or substitutions of meat from a different species is a significant dietary issue, particularly for individuals with allergies or those of a certain religious conviction [3,4]. It is thus prudent to develop techniques in authenticating meat products as a means of ensuring safe trade and ethics [2,3].

Currently, many methods have been developed for the means of food fraud detection, including microscopic, spectroscopic (NMR, FTIR), and DNA-based techniques (PCR) [2,5,6,7]. Indeed, amidst these techniques, biomarkers identification by means of omics technology allows such quantification and distinction at a molecular level [8,9]. In fact, the significant popularity and application of these technologies in resolving food compositions in general at a high resolution has developed an entire field of “foodomics” to identify molecular traits pertaining to the production and processing of these complex mixtures [10,11].

Interestingly, within the available omics technologies in the analysis of meat from different species, the use of glycomics remains relatively unknown. Considering the multiple diverse glycan structures that can arise as a result of the complex protein glycosylation pathways, it is likely that significant structural differences in glycan compositions can be detected between samples derived from different animal species [12,13]. In particular, studies have found at least six different N-glycan structural differences between duck and meat samples as a means of differentiating these two species [13]. Additionally, the types of O-glycan structures in meat samples are not presently known, and thus the discrimination patterns between different meat samples using an O-glycan analysis have also not been resolved. With recent advances in deciphering glycan structures, such as a combined fluorescence-based quantitation with an LC-MS technique (LC-FLR-MS) in resolving N-glycan structures at great sensitivity [14,15], as well as novel methods in releasing O-glycans with free reducing-end aldehydes for O-glycan analysis [16,17,18], a deep and total structural analysis of both N-linked and O-linked glycans can be performed to characterise samples with a degree of resolution that previously would not have been possible.

In this study, we demonstrate the use of this approach to determine the diverse O-linked and N-linked glycan profiles of three meat samples (chicken, beef, and pork). In the O-glycan analysis, we find a clear difference in the distinct structures between the different meat samples. The abundance and diversity of each O-glycan structure appears to be significantly dissimilar, suggesting well-defined O-glycosylation patterns between meat samples derived from different animal species. In the N-glycan analysis, our high-resolution measurements have identified the presence of up to 17 different N-glycan structures in the meat samples, which is an increase by a factor of two compared to the previously carried out analysis [13]. The individual glycan structures, as well as the total glycosylation attributes, are clearly distinguishable between the different meat samples. Finally, a principal component analysis (PCA) is also performed with all O-glycan and N-glycan structures within the meat samples, revealing straightforward discrimination between the samples, and thus the potential use of such integrative glycomic approaches for the high-throughput authentication of meat samples in the future [19].

## 2. Materials and Methods

### 2.1. Meat Lysis and Protein Extraction

Meat samples for each species were purchased and processed within the same day. Chicken samples were isolated from peroneus longus, whilst pork and beef samples were isolated from extensor carpis radialis. Fat and connective tissues were trimmed off from the meat. Pea-sized meat samples were snap-frozen using nitrogen and were minced to homogeneity using a pestle and mortar. Approximately 150 μg of each homogenised meat sample was lysed in 800 μL of T-PER tissue protein extraction reagent supplemented with protease inhibitor (1:100, both from ThermoFisher, UK) for 10 min on a rotary shaker. After this, the samples were centrifuged at 10,000× *g* for 5 min and the supernatant was collected. Proteins extracted were stored at −80 °C before being subjected to the O- and N-glycan analysis workflow.

### 2.2. Release and Permethylation of O-Glycans

O-glycans were released from 100 µg of meat samples by adding 200 µL of (0.5 M) sodium borohydride in 0.05 M potassium hydroxide and incubating in a 50 °C oven for 16 h. The reaction was terminated by adding glacial acetic acid dropwise followed by a clean-up using Dowex 50W-X8(H) 50–100 mesh resin chromatography. The samples were loaded onto the pre-prepared Dowex resin column and the flowthrough was collected in a glass tube. The O-glycans were eluted using 5 mL of 5% acetic acid and combined with the flowthrough. Eluted O-glycans were evaporated to dryness using a nitrogen sample concentrator. Then, 500 µL of 10% acetic acid in methanol was added and dried to remove borate (repeated five times). Sodium hydroxide dissolved in dimethyl sulfoxide and iodomethane were added to the dried glycan samples in glass tubes. The reaction was allowed to proceed under rotation at 30 rpm for about 3 h. Next, 1 mL of deionised water was added dropwise to quench the reaction. After 2 mL of chloroform was added, the mixture was mixed thoroughly. After allowing the mixture to separate into 2 layers, the upper aqueous layer was removed. Deionised water was added to the chloroform layer and this step of mixing and removal of aqueous layer was repeated several times until the chloroform layer was clear. The chloroform layer was then evaporated to dryness using a nitrogen sample concentrator.

### 2.3. Sep-Pak Separation of Permethylated Glycans

C18 Sep-Pak^®^ cartridge (Water Corporation, Milford, MA, USA) was primed sequentially with 5 mL methanol, 5 mL deionised water, 5 mL acetonitrile and 5 mL deionised water. The dried permethylated sample was redissolved in 200 μL of 50% methanol and loaded to the Sep-Pak^®^ cartridge. Elution was carried out by adding 2 mL of 15, 35, 50 and 75% acetonitrile in water (*v*/*v*). Each elution fraction was collected and evaporated to dryness using a SpeedVac.

### 2.4. Mass Spectrometry Analysis of O-Glycans

Permethylated O-glycans from the 35% and 50% fractions were combined and reconstituted in 100 µL of 80% methanol with 0.1% formic acid. Then, 10 μL of reconstituted released O-glycans were injected into Agilent 1290 infinity LC system coupled to an Agilent 6550 iFunel qTOF mass spectrometer (Agilent Technologies, Santa Clara, CA, USA). O-glycans were separated using an Agilent Zorbax Eclipse Plus C18 RRHD column (1.8 μm, 2.1 mm × 50 mm) at 500 μL/min, with an elution gradient of 3 to 10%, 10 to 40%, 40 to 70%, and 70 to 90% of 0.1% formic acid in acetonitrile (ACN, mobile phase B) at 0 to 10 min, 10 to 25 min, 25 to 30 min, and 30 to 38 min, respectively. For mobile phase A, 0.1% formic acid in water was used. The column was flushed with 90% mobile phase B for 12 min before re-equilibrating with 3% mobile phase B for 15 min.

Mass spectra were acquired in positive ion mode over a mass range of *m*/*z* 100–2000 with an acquisition rate of 1 Hz. The following parameters were used for the acquisition: drying gas temperature 150 °C at 12 L/min, sheath gas temperature 300 °C at 12 L/min, nebuliser pressure at 45 psi and capillary voltage at 2500 V. Mass correction was enabled using an infused calibrant solution with a reference mass of *m*/*z* 121.0873 and 922.0098.

### 2.5. O-Glycan Assignment

LC-MS data were processed using Molecular Feature Extractor (MFE) algorithm of MassHunter Qualitative Analysis Software (version B.06.00 Build 6.0.633.10 SP1, Agilent Technologies, Santa Clara, CA, USA). A permethylated mass list was generated based on the neutral masses of O-glycans found on the GlycoStore and Consortium for Functional Glycomics database [20,21]. This list with a mass filter of 10 ppm was used to search the LC-MS data. Mass peaks were filtered with a peak height of at least 100 counts and resolved into individual ion species. Using a Glycans Isotopic distribution model, charge state of a maximum of 3 and retention time, all ion species with singly and doubly protonated ions and their sodium adducted ions associated with a single compound were summed together. The neutral compound mass was then calculated and a list of all compound peaks in the samples and standards were generated with relative abundances depicted by chromatographic peak areas.

Targeted tandem MS was acquired in positive ion mode over a mass range of *m*/*z* 100–2000 with an acquisition time of 1.5 Hz. A targeted mass list was generated based on the desired MFE compounds found on the samples for MS/MS analysis. The precursor masses of interest, along with its charge state, retention time and peak width were indicated. The isolation width used was medium (~4 *m*/*z*) and the collision energy (CE) used for each precursor compound were automatically calculated by the acquisition software based on the following equation:CE (eV) = 3 (*m*/*z* ÷ 100) − 4.8(1)

The targeted tandem MS data were processed using MFE algorithm with the same settings used for searching LC-MS data and the MS/MS spectrum were extracted from each of the targeted compounds.

### 2.6. N-Glycan Release and Labelling

N-glycans of meat were released and labelled using GlycoWorks RapiFluor-MS (RFMS) N-glycan kit (Waters Corporation, Milford, MA, USA) according to the manufacturer’s protocol. Briefly, 15 μg of dried protein extracted from meat was reconstituted in 22.8 μL of LCMS grade water and 6 μL of 5% RapiGest solution (final concentration 0.01% RapiGest, Waters Corporation, Milford, MA, USA). The solution was incubated at 95 °C for 5 min to denature the protein extracted from meat. N-glycans were released enzymatically by adding 600 U of recombinant PNGase F (New England Biolabs, Ipswich, MA, USA) followed by 10 min incubation at 55 °C. Released N-glycans were labelled with 12 μL of the RapiFluor-MS Reagent Solution (fluorescence label, 0.07 mg/μL in anhydrous dimethyformamide (DMF), Waters Corporation, Milford, MA, USA) at room temperature for 10 min. The solution was diluted in 358 μL of ACN, followed by clean-up using a GlycoWorks HILIC μElution Plate (Waters Corporation, Milford, MA, USA). Isolated released N-glycans were dried and reconstituted in 9 μL of LCMS grade water, 10 μL of DMF, and 21 μL of ACN sequentially for LC-MS analysis.

### 2.7. Liquid Chromatography-Mass Spectrometry Analysis of RFMS-Labelled N-Glycan

Released N-glycans were analysed as previously described [22]. First, 10 μL of reconstituted released N-glycans were injected into an ACQUITY H-Class UPLC (Waters Corporation, Milford, MA, USA) coupled to a SYNAPT XS mass spectrometer (Waters Corporation, Milford, MA, USA). Samples were separated using an ACQUITY UPLC Glycan BEH amide column (130 A, 1.7 μm, 2.1 mm × 150 mm, Waters Corporation, Milford, MA, USA) at 60 °C and 400 μL/min, with a 40 min gradient from 25 to 49% of 50 mM Ammonium Formate (mobile phase A). As mobile phase B, 100% ACN was used. RFMS-labelled glycans were excited at 265 nm and measured at 425 nm with an ACQUITY UPLC FLR detector (Waters Corporation, Milford, MA, USA). The MS1 profile scans of *m*/*z* 400–2000 were acquired using the SYNAPT XS in positive mode with an acquisition rate of 1 Hz. The electrospray ionisation capillary voltage was set at 1.8 kV, cone voltage at 30 V, desolvation gas flow at 850 L/h, and ion source temperature and desolvation temperature were kept at 120 °C and 350 °C, respectively. Leucine Enkephalin (Waters Corporation, Milford, MA, USA) was used as the LockSpray compound for real-time mass correction. RapiFluor-MS Dextran Calibration ladder (Waters Corporation, Milford, MA, USA) was also injected into LC-MS to calibrate the retention time of sample peaks. The retention times were normalised using the dextran calibration curve to Glucose Units (GU).

### 2.8. N-Glycan Assignment

Released N-glycans were analysed using the UNIFI Scientific Information System (Version 1.8, Waters Corporation, Milford, MA, USA). Fluorescence peaks were integrated manually using the UNIFI Scientific Information System and relative quantitation of peaks was obtained by area-under-curve measurements followed by normalisation to the total area. Glycan assignment was carried out by matching neutral mass and/or Glucose Units (GU) of each peak to the modified “N-glycan 309 mammalian no sodium” database available in the Byonic software.

### 2.9. Statistical Analyses

Data are presented as mean ± standard error of the mean. Statistical analyses were performed by Student’s *t*-test or one-way ANOVA with Tukey’s post hoc test using GraphPad Prism 8 (GraphPAD Software Inc., San Diego, CA, USA). Multiple comparisons were performed between chicken, pork, and beef in ANOVA tests. O-glycan and N-glycan relative abundance was analysed using principal component analysis (PCA). The criterion for significance was *p* < 0.05 (* *p* < 0.05; ** *p* < 0.01; *** *p* < 0.001; **** *p* < 0.0001).

## 3. Results

### 3.1. Framework for an Integrated Glycomic Study of Meat Samples from Different Species

The overall glycomic-based workflow consists of: (i) an experimental approach in extracting and isolating glycan structures from meat samples, (ii) a quantitative glycomic analysis of the abundance of both N-glycans and O-glycans, and finally (iii) the computation of the principal component analysis (PCA) to successfully discriminate glycans belonging to the different meat samples (Figure 1).

In the first step, whole meat samples are grounded and subjected to lysis using T-PER^TM^ Tissue Protein Extraction Reagent. Subsequently, a centrifuge step is employed to pellet the tissue debris, and the final supernatant product contains all proteins successfully extracted from the meat samples, including the glycoproteins of interest. In the next step, the same samples are split for N-glycan and O-glycan analysis. In the O-glycan analysis, O-glycans are released and permethylated before the LC-MS analysis, while in N-glycans, they are released and labelled with Rapifluor-MS (RFMS) before an UPLC-FLR-MS analysis. Both workflows allow the differentiation and quantification of the abundance of distinct N-glycans and O-glycans. Finally, these results are pooled together for a PCA analysis which allows the novel discrimination of the different meat samples on a molecular level (Appendix A).

### 3.2. O-Glycan Characterisation of Meat

We demonstrate the application of this workflow by applying the above workflow to three types of meat samples, namely chicken, pork, and beef. The extracted proteins were first analysed based on their O-glycan profiles. O-glycan structures, in particular, have been shown to exhibit a diverse glycosylation pattern in eukaryotes, owing to the many biosynthetic pathways [23]. In the case of the three meat samples, we observe the presence of four distinct O-glycan structures found through the analysis of released permethylated O-glycans. Through the combined analysis of the retention time and MS2 fragment data of the O-glycan standards, two of the most abundant glycans were identified as Gal-GalNAc and NeuAc-Gal-GalNAc (Figure 2 and Figure 3A,B). Gal-GalNAc, in particular, was observed to be the most abundant structure in all samples, though its relative abundance ranged from 74.7 ± 0.6% in pork meat to 45.7 ± 3.3% in chicken meat (Figure 3A, Table 1). This significant abundance corroborates previous observations of Gal-GalNAc as one of the core, and thus most abundant, O-glycan structures that can be found, particularly in mammals [24,25]. On the other hand, the relative abundance of NeuAc-Gal-GalNAc was observed to range from 10–25% instead, depending on the sample. The presence of two other distinct glycans was also observed, though their exact linkage could not be ascertained. These two structures, labelled instead through their chemical compositions as Hex-HexNAc (RT~15.5 min), and Hex(NeuAc)HexNAc (RT~18.8 min), respectively, was observed to be more abundant in the chicken meat samples than the other animal samples (Figure 2 and Figure 3C,D).

Furthermore, a quantitative comparison of the relative abundances of all four O-glycans was made across all three meat samples (Figure 3A–D, Table 1). ANOVA test indicated that the relative abundances of Gal-GalNAc and Hex-HexNAc (by chemical composition) in chicken, pork, and beef meat samples are significantly different from each other (Figure 3A,C). An interesting trend was observed in the case of sialylated O-glycans within the meat samples. In particular, chicken meat samples were observed to contain a significant amount of Hex(NeuAc)HexNAc (by chemical composition) which was otherwise undetected in the beef and pork samples (Figure 3D). However, in the case of the other sialylated O-glycan NeuAc-Gal-GalNAc, chicken meat samples had the lowest relative abundance as compared to the other two samples (Figure 3B, 11.3 ± 1.9% versus 25.3 ± 0.6% and 23.1 ± 3.6%, respectively, *p* < 0.05, *p*< 0.01). Indeed, the combined quantification of all sialylated glycans (Hex(NeuAc)HexNAc and NeuAc-Gal-GalNAc) yield no statistically significant difference between species (Figure 3E). This highlights the importance of high resolution glycomics in characterising individual O-glycan structures for the proper distinction between meat samples of different animal origins.

### 3.3. N-Glycan Characterisation of Meat

We also performed a full characterisation of the N-glycan structures of the extracted proteins from the three meat samples (Figure 4). We released N-glycans using the enzyme PNGaseF and labelled them with RFMS fluorescence tag before subjecting them to FLR-LC-MS workflow. Interestingly, from the FLR chromatogram, which reflects the abundance of the fluorescently labelled N-glycans, we can observe distinct overall N-glycomic signatures between the samples. For instance, in the case of the chicken meat sample, an even distribution of peaks was observed (Figure 4A), whilst distribution of peaks from pork and beef meat samples were skewed towards those with higher retention time, which also corresponds to higher neutral mass (Figure 4B,C). In particular, the overall N-glycome of the pork sample appeared to be less heterogeneous, with one major peak observed at around 20.8 min. N-glycan compositions were further confirmed with neutral mass and/or glucose unit (GU) and labelled in the representative FLR chromatogram (Figure 4). We subsequently quantified the relative abundance of each N-glycan structure between the meat samples (Table 2). In each sample, up to 17 different glycan structures could be identified, which is significantly higher than that reported from previous studies [13]. We note that other rare, low-abundant glycan structures may also be present in these samples, whose signal is masked by the relatively much more abundant glycan structures. These N-glycans of these samples were grouped based on their glycosylation attributes—fucosylation, sialylation, galactosylation, and the presence of high mannose (Figure 5). We observed that the pork meat samples contained the highest abundance of fucosylated N-glycan structures (Figure 5A, *p* < 0.0001 and *p* < 0.05, for comparison between pork and chicken or beef, respectively), followed by beef and chicken samples (Figure 5A, *p* < 0.001, for comparison between chicken and beef). Chicken meat samples contained the lowest relative abundances of sialylated and galactosylated N-glycans compared to pork and beef meat samples (sialylation: Figure 5B, *p* < 0.001 and *p* < 0.01, for comparison between chicken and pork or beef, respectively; galactosylation: Figure 5C, *p* < 0.001, for both comparison between chicken and pork or beef), with no statistically significant difference between pork and beef meat samples. In contrast, the chicken meat sample had the highest amount of N-glycans with high mannose structure (Figure 5D, *p* < 0.001, for both comparison between chicken and pork or beef), while no statistically significant difference was found between pork and beef samples.

The relative abundances of two predominant sialic acids on N-glycans, Neu5Ac and Neu5Gc were also characterised, considering the prominent role of sialic acids in multiple biological functions [26]. In particular, the uptake of Neu5Gc from red meat has been shown to trigger inflammatory response and cancer development [27]. Our results show that pork samples contained significantly higher relative abundance of Neu5Ac than chicken and beef samples, with no difference between chicken and beef samples (Figure 5E, *p* < 0.001, *p* < 0.01, for comparison between pork and chicken or beef, respectively). On the other hand, no Neu5Gc was detected in chicken, with 4.8 ± 0.3% and 34.4 ± 1.9% of Neu5Gc detected in pork and beef, respectively (Figure 5F). More than half of the total sialic acid content (Neu5Ac and Neu5Gc) found in beef was Neu5Gc, compared to only a significantly small proportion of Neu5Gc detected in pork (Figure 5G, 58.2 ± 1.8% versus 6.9 ± 0.4%). The ratio of Neu5Gc to total sialic acid in all three meats agrees with previously published work that analysed free sialic acid using HPLC method [27].

Our high resolution glycomic workflow also allows the full characterisation of N-glycans with α-galactose (galactose-α-1,3-galactose). It is important to characterise the abundance of α-galactose in meat samples because it is implicated in alpha-gal syndrome (also known as red meat allergy)—a potentially life-threatening allergy to red meat [28]. The highest relative abundance of α-galactose amongst the meat samples was detected in the beef meat sample, followed by a moderate amount found in the pork sample, and an undetectable amount in chicken meat samples (Figure 5H, 35.8 ± 1.8%, 5.1 ± 0.8%, and 0%, respectively). This agrees with our understanding that α-galactose is found more abundantly in red meat than in white meat [29].

### 3.4. PCA Analysis

Our characterisation of the N-linked and O-linked glycans of the three different meat samples shows the distinct differences in their relative abundances, suggesting a unique overall glycomic profile of each sample that can be distinguished with the integrative glycomic workflow. In order to achieve a unified glycome analysis of both N-linked and O-linked glycans, the datasets were pooled together and subjected to principal component analysis (PCA). The dataset was transformed into two principal components—PC1 and PC2,—which explained percentages of variance, which were 57.4% and 25.8%, respectively. The reliability of this PCA analysis can be observed by more than 82% of the total variance that can be accounted for by the first two PCs. The score plot of PC1 and PC2 of each meat allowed visual discrimination of different species (Figure 6). This analysis demonstrates, with strong statistical significance, the well-defined glycomic characteristics of meat samples pertaining to different animal species, and the strength of the overall integrated glycomic approach in profiling meat samples.

## 4. Discussion

Current glycomic approaches are extensively applied for the use as biomarkers in different areas, particularly in medical and biotechnology fields [30,31,32]. In this study, we have described in detail the extended use of an integrated glycomics approach to characterise meat samples in terms of their O-linked and N-linked glycan structures. The abundances of individual glycan types are significantly different both in terms of the O-linked as well as N-linked glycans. Previous studies have investigated the O-linked glycans found in food, such as bovine whey protein product and mucins from salmon and chicken [33,34,35]. However, to the best of our knowledge, O-linked glycan characterisation has not been employed in meat profiling. The novel discovery of the main types of O-glycan structures found in meat samples shows different molecular signatures pertaining to each species, and an orthogonal measurement in glycomics for meat differentiation purposes. Similarly, the identification of a diverse number of N-glycan structures in each sample shows the diversity of mechanisms which each species undergoes for glycosylation, resulting in their distinct structural differences. Despite the variety of N-glycan structures that can be present, it is interesting that meat samples of each animal species possess a distinct subset of these structures. This suggests a high degree of specificity, especially in the glycoenzymes involved in the synthesis of these glycans [8].

It is worth noting that meat tissues were isolated only from one region of the animal (peroneus longus from chicken samples, and extensor carpis radialis from the pork and beef samples). As glycosylation is context- and tissue-specific, we anticipate potential differences in the glycan profile of different tissues belonging to the same animal [36,37]. By systematically comparing the differences between tissue sources, and between animals, future studies can further reveal the potential of glycomics in its versatile applicability in different types of food samples. In addition, this study has not characterised the glycan profile of all animal species. It has been observed that meat species adulteration can occur between meat samples of closely related species such as horse meat and beef [1,38]. Thus, further studies investigating the glycome differences between meat samples derived from closely related species are warranted.

The meat industry presents other major challenges such as fraudulent labelling of geographical origin and production system (e.g., organic vs. non-organic) of the meat samples. Amongst the many analytical techniques available in the meat industry, there are only a few capable of performing the authentication of geographical origin and production system. For instance, DNA-based methods such as polymerase chain reactions and genomics rely on a specific DNA sequence or taxonomic marker to differentiate the species of meat, but they are unable to detect fraudulent labels of geographical origin and production system. Since dietary patterns, lifestyle and environmental changes are known to affect protein glycosylation in humans, animals would likely also undergo glycan changes. As such, such changes in glycan profile could be exploited in the authentication of geographical origin and production system. The use of glycomic techniques in meat species profiling presented in this paper serves as a proof-of-concept and its use in the authentication of other meat product features should be investigated. Importantly, glycoproteomic approaches harnessing both proteomic and glycomic potentials could also be a promising tool to further the molecular characterisation of meat samples.

Considering the traction of cultured meat products (growing cells to generate meat-like tissue structures in the laboratory) as an alternative protein source for human consumption, the glycoprofile workflow described in this study can also be used in establishing the critical quality attributes (CQA) of cultured meat products [39,40,41]. Given the considerable risk of cell line contamination and product adulteration in this growing industry, such techniques can help to establish a stringent quality control of these cultured meat samples in the future [42,43]. With the advent of FDA-approved genetically modified pigs with α-galactose for human consumption, our glycomic techniques can also be used to monitor the controlled manipulation of cultured meat [44]. This includes their modification in consideration of unwanted glycans such as α-galactose and Neu5Gc for health and safety reasons.

While more studies are warranted to investigate how these glycomic signatures may differ based on breed genetic compositions, parts from which the meat samples were obtained, and other extrinsic factors (e.g., feed intake, growth conditions, regional differences), it is evident that the use of O-linked and N-linked glycome profiling allows the successful differentiation between different meat samples, and as a proof-of-concept paves the way for a new high-throughput and robust approach in quantifying meat adulteration. This may be particularly advantageous over other approaches as N-glycan profiling is unaffected by the harsh processing of meat (heat-induced treatments) that can degrade DNA and affect the accuracy of genomic approaches [8]. Given its highly quantitative and efficient procedure, the adoption of such glycome profiling approaches provides a powerful future alternative technique to traditional methods in meat identification and authentication.

## Figures and Tables

**Figure 1 foods-11-01952-f001:**
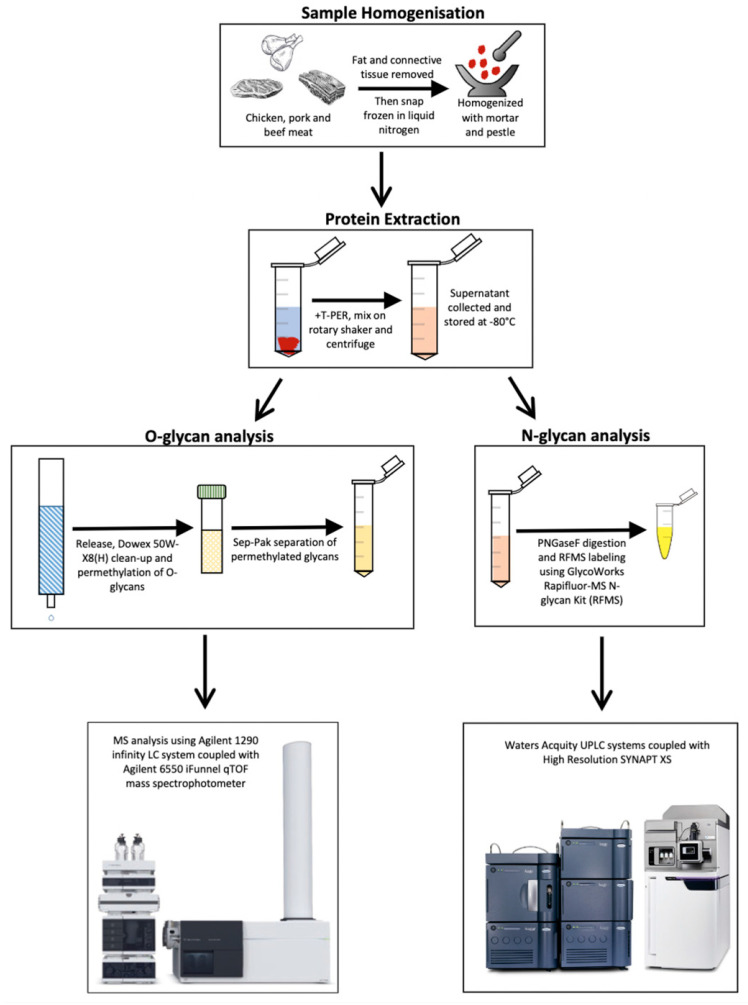
Glycomics-based workflow for the characterisation and profiling of meat samples. Samples are first processed for the extraction of proteins, which undergo separate procedures for the release of N-linked and O-linked glycans. These glycans are finally quantitated via a mass spectrometry-based analysis. Permethylated O-glycans are quantitated through the mass abundance of the ions whilst N-glycans are quantitated via the fluorescence of the RFMS label.

**Figure 2 foods-11-01952-f002:**
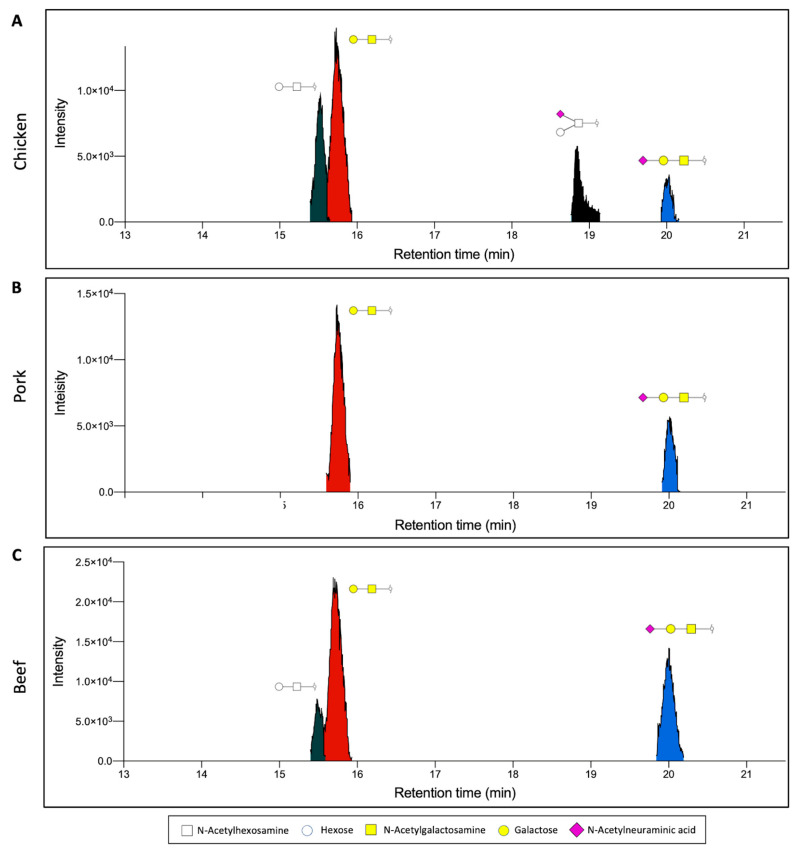
Representative extracted ion chromatogram of permethylated O-glycans extracted from (**A**) chicken, (**B**) pork, and (**C**) beef. Gal-GalNAc and NeuAc-Gal-GalNAc were detected in all species, whilst Hex-HexNAc was detected only in chicken and beef and Hex(NeuAc)HexNAc was detected exclusively in chicken. (White square, N-acetylhexosamine; white circle, Hexose; yellow square, N-acetylgalactosamine; yellow circle, galactose; purple diamond, N-acetylneuraminic acid).

**Figure 3 foods-11-01952-f003:**
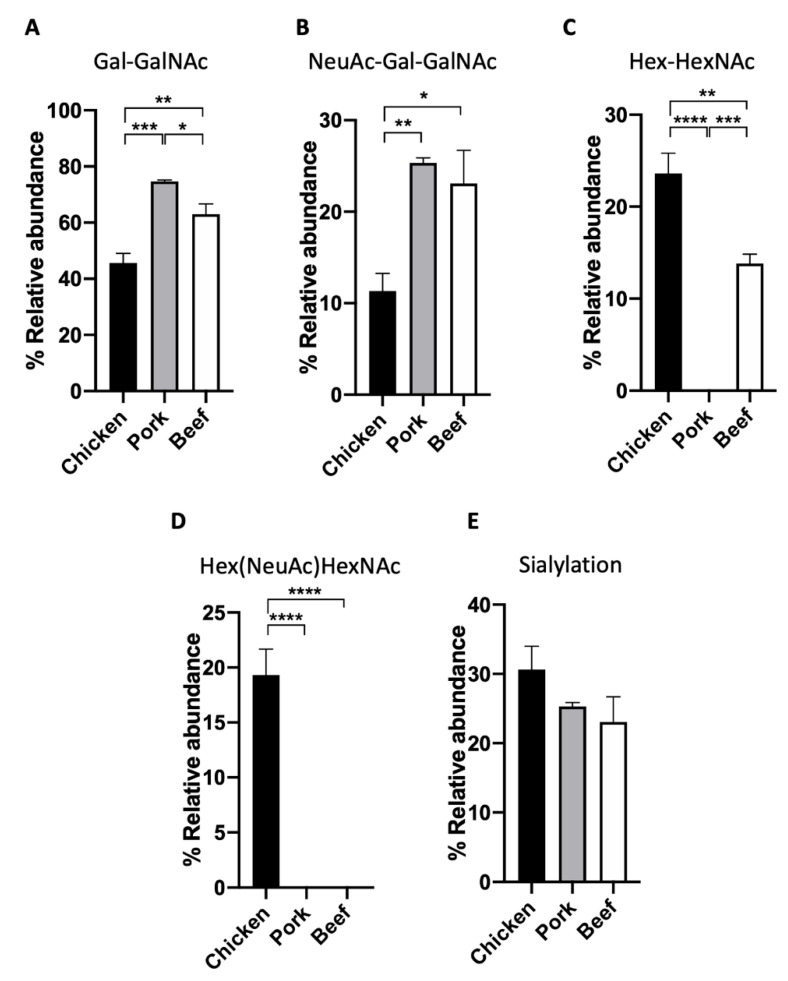
The relative abundance of permethylated O-glycans in meat samples. (**A**) Gal-GalNAc, (**B**) NeuAc-Gal-GalNAc, (**C**) Hex-HexNAc, (**D**) Hex(NeuAc)HexNAc, and (**E**) sialylation of chicken, pork and beef. (One-way ANOVA followed by Tukey’s post hoc test, multiple comparisons were performed between chicken, pork, and beef, *n* = 4; * *p* < 0.05; ** *p* < 0.01; *** *p* < 0.001; **** *p* < 0.0001).

**Figure 4 foods-11-01952-f004:**
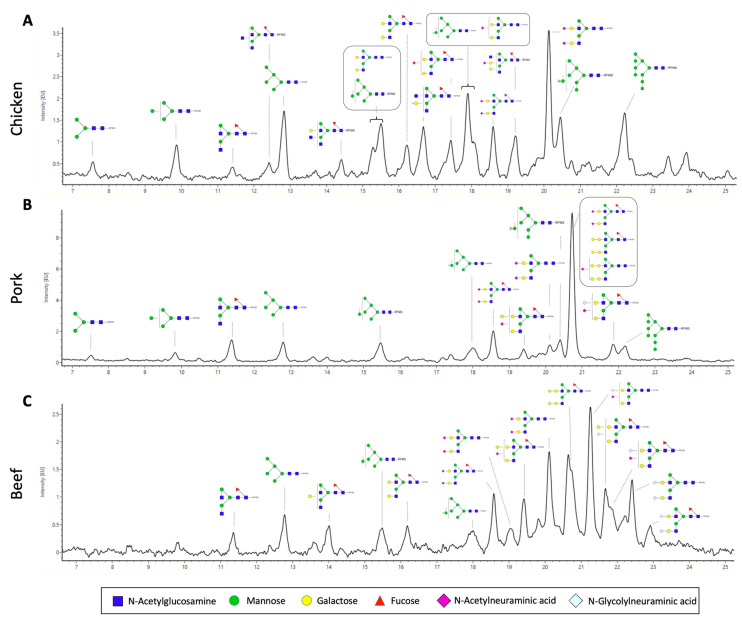
Representative fluorescence (FLR) chromatograms of RFMS-labelled released N-glycans from (**A**) chicken, (**B**) pork, and (**C**) beef. (Blue square, N-Acetylglycosamine; green circle, mannose; yellow circle, galactose; red triangle, fucose; purple diamond, N-acetylneuraminic acid; light blue, N-Glycolylneuraminic acid.) Glycan images represent compositions and linkage type is not determined. (Samples were separated using an ACQUITY UPLC Glycan BEH amide column).

**Figure 5 foods-11-01952-f005:**
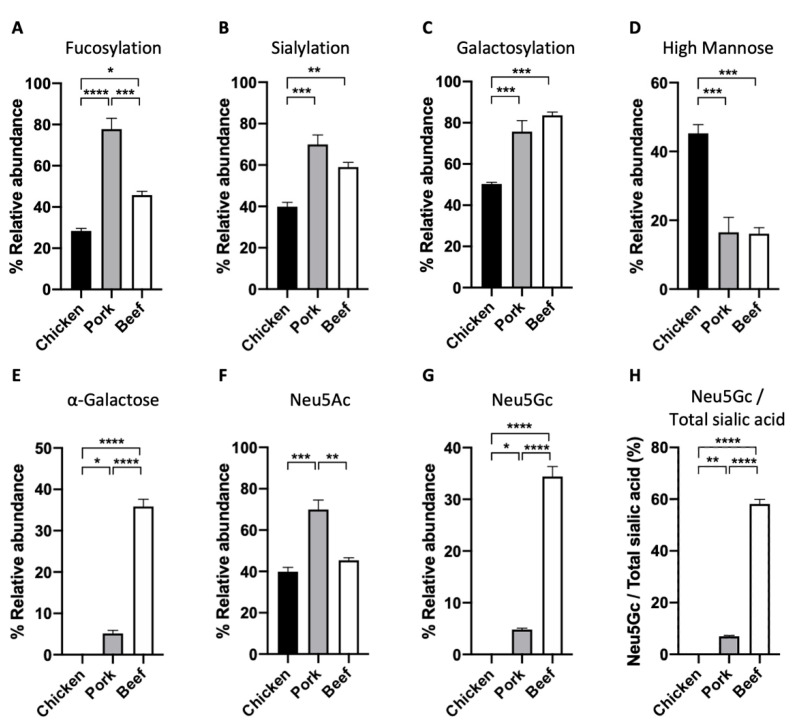
The relative abundance of (**A**) core fucosylation, (**B**) sialylation, (**C**) galactosylation, (**D**) high mannose, (**E**) α-galactose, (**F**) N-acetylneuraminic acid (Neu5Ac), and (**G**) N-glycolylneuraminic acid (Neu5Gc) measured in N-glycans of chicken, pork, and beef. (**H**) Ratio of Neu5Gc and total sialic acid was also calculated. (One-way ANOVA followed by Tukey’s post hoc test, *n* = 4 independent biological replicates; * *p* < 0.05; ** *p* < 0.01; *** *p* < 0.001; **** *p* < 0.0001).

**Figure 6 foods-11-01952-f006:**
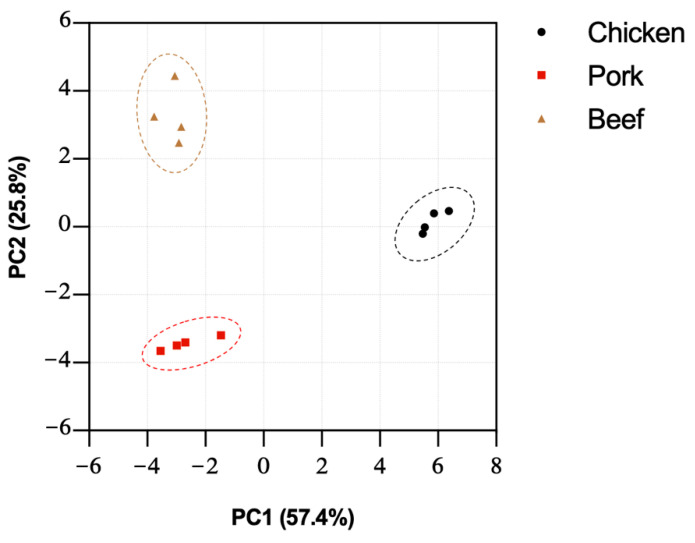
Principal component analysis was performed on relative abundances of O- and N-glycans of chicken, pork, and beef. A clear separation was observed between meat from different species. Each dot represents an independent biological replicate.

**Table 1 foods-11-01952-t001:** Summary of relative abundances of O-glycans in percentage detected in chicken, pork, and beef. (±, standard error of mean; ND, not detected).

	Chicken (%)	Pork (%)	Beef (%)
Hex-HexNAc	23.6 ± 2.2	ND	13.8 ± 1.0
Gal-GalNAc	45.7 ± 3.3	74.7 ± 0.6	63.1 ± 3.6
Hex(NeuAc)HexNAc	19.3 ± 2.4	ND	ND
NeuAc-Gal-GalNAc	11.3 ± 1.9	25.3 ± 0.6	23.1 ± 3.6

**Table 2 foods-11-01952-t002:** Summary of relative abundances of N-glycans detected in chicken, pork, and beef.

	Chicken (%)	Pork (%)	Beef (%)
HexNAc(2)Hex(3)	1.0 ± 0.4	0.3 ± 0.3	ND
HexNAc(2)Hex(4)	2.9 ± 0.8	0.5 ± 0.5	1.0 ± 1.0
HexNAc(4)Hex(3)Fuc(1)	1.1 ± 0.2	6.3 ± 0.2	0.8 ± 0.5
HexNAc(2)Hex(5)	9.3 ± 0.2	4.3 ± 0.5	5.8 ± 0.3
HexNAc(4)Hex(4)Fuc(1)	ND	1.2 ± 0.4	1.8 ± 1.4
HexNAc(5)Hex(4)Fuc(1)	1.6 ± 0.1	ND	ND
HexNAc(4)Hex(5)	2.0 ± 0.2	0.3 ± 0.1	ND
HexNAc(2)Hex(6)	8.1 ± 0.8	4.5 ± 1.0	4.1 ± 1.4
HexNAc(4)Hex(5)Fuc(1)	2.2 ± 0.4	ND	2.5 ± 0.5
HexNAc(5)Hex(5)Fuc(1)	5.9 ± 0.5	ND	ND
HexNAc(4)Hex(5)NeuAc(1)	9.4 ± 0.5	ND	ND
HexNAc(4)Hex(5)Fuc(1)NeuAc(1)	6.2 ± 0.6	0.2 ± 0.2	ND
HexNAc(2)Hex(7)	6.6 ± 0.6	3.9 ± 0.6	4.1 ± 0.9
HexNAc(4)Hex(5)Fuc(1)NeuAc(2)	4.3 ± 0.5	60.7 ± 6.3	4.6 ± 0.1
HexNAc(5)Hex(5)Fuc(1)NeuAc(1)	6.0 ± 0.3	ND	ND
HexNAc(4)Hex(6)Fuc(1)NeuAc(1)	ND	1.2 ± 0.5	8.1 ± 0.7
HexNAc(4)Hex(5)NeuAc(2)	11.9 ± 1.0	2.3 ± 0.2	9.7 ± 1.0
HexNAc(2)Hex(8)	10.8 ± 1.1	2.2 ± 1.4	0.6 ± 0.1
HexNAc(4)Hex(7)Fuc(1)	ND	3.3 ± 0.3	17.8 ± 1.3
HexNAc(4)Hex(6)Fuc(1)NeuGc(1)	ND	ND	4.2 ± 0.4
HexNAc(4)Hex(7)NeuAc(1)	ND	1.1 ± 0.1	2.3 ± 0.2
HexNAc(4)Hex(5)NeuAc(1)NeuGc(1)	ND	ND	18.5 ± 1.4
HexNAc(4)Hex(5)Fuc(1)NeuAc(1)NeuGc(1)	ND	4.9 ± 0.3	2.2 ± 1.0
HexNAc(2)Hex(9)	9.2 ± 0.7	2.3 ± 1.4	ND
HexNAc(4)Hex(5)NeuGc(2)	ND	ND	5.8 ± 0.4
HexNAc(4)Hex(5)Fuc(1)NeuGc(2)	ND	ND	2.5 ± 1.0

Hex, hexose sugars; HexNAc, N-acetylhexosamine; Fuc, fucose; NeuAc, N-acetylneuraminic acid; NeuGc, N-glycolylneuraminic acid. The number beside each sugar refers to the number of such molecules present in that structure. (±, standard error of mean; ND, not detected).

## Data Availability

The data presented in this study are available on request from the corresponding author.

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
