# Peer review of "An Integrative Glycomic Approach for Quantitative Meat Species Profiling"

_foods, 2022, doi:10.3390/foods11131952_

Round 1

Reviewer 1 Report

The authors posit a novel strategy to tackle the issues of food fraud, in particular the issue of species differences' being mislabeled either as an act of deliberate fraud or improper provenance. This issue has multiple interested parties, including consumer reports activists, animal welfare agents, religious groups, and those concerned for medical reasons (eg allergies such as alpha-gal). 

I would recommend that the authors either address the following issues directly or adjust language to suit: 

- the authors state that multiple methods for food fraud detection exist, but that the use of glycomics is a newcomer to the field. spectroscopic, microscopic, and proteomics methods are all much faster or more established methods. Can you provide reasons for what what advantages glycomics would give over these established methods? Proteomics is a common method for this field - should glycoproteomics be mentioned?

-flesh out the following more in the discussion: the authors provide replicate samples for glycomics analysis from three species but from only tissue source. Glycosylation as a PTM is heavily "context-specific" so it would be expected that glycosylation from chicken breast would be different that that of chicken thigh. 

- flesh out the following more in the discussion: the authors provide samples from chicken, beef, and pork which are the three most common animal meat sources, and show statistical differences between them. However, many issues of food fraud do not involve chicken being confused for beef, they involve horse being confused for beef. 

-The authors use two different workflows and MS systems to analyze O-glycans versus that of N-glycans. This is somewhat understandable since no reliable O-glycanase is commercially available, while PNGase F/A is a standard workflow for N-glycomics. What is less clear from this report is why the authors decided to permethylate the beta-eliminated O-glycans but use a commercial kit for the N-glycans? Why not permethylate both? Both methods are appropriate strategies but few labs would consider attempting both, nor would they likely have both instruments available. 

Reviewer 2 Report

Foods Chia et al

The authors report on N- and O-glycan profiling of meat from different vertebrate species in order to uncover differences which may be relevant to detection of food fraud. While it is true that a rapid method for such analyses is required, the basic underlying data seems to be a severe underestimate of the glycomic complexity in the samples. A couple of grammatical errors were obvious, thus the authors also need to re-read the manuscript carefully. The overall idea that NeuGc and alpha-Gal are potential ‘fraud markers’ is fine. I have previously reviewed this manuscript for another journal and only some minor changes were made – thus, many of my previous comments are included below.

Introduction – the authors discuss the need for a deep and total analysis, but then find only maximally 17 different N-glycan structures. Personally in tissue samples I would be expecting 100 or so structures (including isomers).

Methods – O-glycans – rather than organic extraction, use only of the C18 SPE may be more appropriate as sulfated glycans will not be lost and be also more applicable to a rapid method. Reliance on databases will also limit the output – was ‘de novo’ manual assignment used? Beta-elimination will also release N-glycans.

N-glycans – how good is the release after 10 minutes at 55 °C? Again, the authors rely on a database which again will limit the output. Also, no MS/MS and no negative-mode MS was performed. Considering the presented chromatograms with (generally) one glycan per peak, I think the authors limit the depth of their analyses. At least in the supplement they should present mass spectra of each fraction.

Title of section “Mass spectrometry analysis O-glycan” missing the word ‘of’ and a plural ‘glycanS’; why is part of this section in italics? Plural glycanS is also missing from the title for the 'release' section.

Results – see my comments regarding the N- and O-glycans.

Figure 2 – the authors major focus in the text is on the O-glycans with the N-glycans being an ‘add-on’ – but NeuGc is absent (seemingly) from the O-glycans, but is present on N-glycans – so it seems that the major ‘differentiation’ of the meats is rather in the N-glycans.

Figures 2 and 4 are of poor quality/resolution.

Figure 4 – state the column type in the figure legend (BEH is normal phase).

Figure 5 – is this core or antennal fucosylation? and these data are all on N-glycans? (not stated).

Parts of the manuscript are highlighted in grey.

Having percentages and deviations to 2 decimal places is an exaggeration of the sensitivity of the methods employed.

Discussion – line 375 – ‘that can manifest’ – rather ‘are manifest’? Alpha-galactose could be written with the Greek letter.

Line 387 – ‘This includes removal of unwanted glycans ..’ to be exact the glycans are not removed, but glycan epitopes are reduced/eliminated (but not in the chemical sense). Removal would mean using a glycosidase treatment of the meat – which is not realistic.

Round 2

Reviewer 2 Report

The authors have resubmitted within two weeks and have made some textual changes, but the basic problem remains that the basic underlying data seems to be a severe underestimate of the glycomic complexity in the samples. Just because others found even fewer structures, does not mean that the authors should not fully apply the power of LC separation in conjunction with MS – which should overcome the suppression of minor glycans.

There are only 5 N-glycan MS shown per sample in the Supplement; there are multiple species in most fractions, especially the Man5; the x-axis is given as mass, which is not the same as m/z; there is an ion at 614 in almost every mass spectrum. I assume these are all doubly-charged ions; more should be annotated and negative mode MS performed. The authors should follow international MIRAGE guidelines for reporting glycomic data.

The O-glycomic data is also extremely rudimentary with only two, three or four structures per sample! It would be better also to remove the grey panel from around each spectrum or are these screenshots? The 19 min peak seems to float in the chicken panel.

The argument that the less common glycans can be ignored is actually incorrect. There is often a tendency for the more species- or tissue-specific glycans to be exactly those of lower abundance!

Lines 124-128 are in italic - this is not a heading, but a continuation of a previous paragraph.
